# Metabolic balancing by miR-276 shapes the mosquito reproductive cycle and *Plasmodium falciparum* development

Lena Lampe[1,3], Marius Jentzsch[1], Sylwia Kierszniowska[2] & Elena A. Levashina [1]*

The blood-feeding behavior of *Anopheles* females delivers essential nutrients for egg development and drives parasite transmission between humans. *Plasmodium* growth is adapted to the vector reproductive cycle, but how changes in the reproductive cycle impact parasite development remains unclear. Here, we show that the bloodmeal-induced miR-276-5p fine-tunes the expression of *branched-chain amino acid transferase* to terminate the reproductive cycle. Silencing of miR-276 prolongs high rates of amino acid (AA) catabolism and increases female fertility, suggesting that timely termination of AA catabolism restricts mosquito investment into reproduction. Prolongation of AA catabolism in *P. falciparum*-infected females also compromises the development of the transmissible sporozoite forms. Our results suggest that *Plasmodium* sporogony exploits the surplus mosquito resources available after reproductive investment and demonstrate the crucial role of the mosquito AA metabolism in within-vector parasite proliferation and malaria transmission.

[1] Vector Biology Unit, Max Planck Institute for Infection Biology, Charitéplatz 1, 10117 Berlin, Germany. [2] metaSysX GmbH, 14476 Potsdam-Golm, Germany. [3] Present address: Physiology and Metabolism Laboratory, Francis Crick Institute, 1 Midland Road, NW1 1AT London, UK. *email: levashina@mpiib-berlin.mpg.de

The hematophagous lifestyle of *Anopheles* females is an efficient reproductive strategy that is exploited by the malaria parasite for its transmission. Mosquitoes mostly feed on carbohydrate-rich plant nectars and only females take blood to kick-start their reproductive cycle. The blood meal provides the female with a nutritional boost in amino acids and lipids that is crucial for egg development. This extreme change in diet triggers massive coordinated metabolic changes in multiple mosquito tissues to ensure robust egg development within a 3-day time window.

The insect fat body is the main nutrient storage organ and plays a central role in the mosquito metabolism. During a pre-blood-feeding anabolic phase, the fat body accumulates nutrients in the form of triacylglycerides (TAGs) and glycogen derived from plant nectars[1–4]. A successful blood feeding massively increases levels of free amino acids (AAs) in the mosquito circulation, that are sensed in the fat body by the target of rapamycin (TOR) pathway[5–7]. Activation of TOR, together with the increasing levels of the steroid hormone 20-hydroxyecdysone (20E), initiate a switch from anabolic to catabolic metabolism[1,2,7]. In the fat body, blood meal-derived AAs feed into the tricarboxylic acid (TCA) cycle to generate energy for nutrient mobilization and are incorporated into major yolk proteins such as lipophorin and vitellogenin[8,9]. Inhibition of nutrient transport, metabolism, or the TOR signaling pathway dramatically reduces egg production[5,10–13]. The mosquito reproductive cycle can be roughly separated into three phases[1,2]: During the early (0–10 h post blood feeding (hpb)) and mid-phase (10–36 hpb), the metabolic program prioritizes the mobilization of nutrients for rapid egg development. During the late phase (36–72 hpb), production of the yolk proteins in the fat body ceases, and transcriptional and metabolic programs return to the anabolic state to replenish consumed glycogen and lipid reserves. The metabolic switch from catabolism to anabolism during the late phase prepares females for the next reproductive cycle. Indeed, prolongation of the reproductive cycle by inhibition of autophagy or silencing of a negative regulator of TOR signaling in the fat body curbs egg production in the second reproductive cycle[14,15]. Therefore, both the induction and termination of the reproductive cycle are essential for female fertility.

Mosquitoes acquire malaria parasites when feeding on blood infected with *P. falciparum* sexual stages that fuse in the mosquito midgut to produce motile ookinetes. The ookinetes traverse the midgut epithelium at 18–24 hpb and transform into oocysts to establish infection at the basal side of the midgut wall. Here, the parasites undergo massive replication and maturation (also called sporogony). Within two weeks of sporogony, the parasite biomass increases dramatically generating up to 1000 sporozoites within each oocyst[16]. Developing oocysts require large amounts of nutrients, such as AAs, sugars, and lipids, for which they rely on the mosquito environment. Therefore, it is not surprising that the mosquito's nutritional status largely correlates with parasite development[17,18]. Inhibition of sugar uptake or lipid access of the *Plasmodium* oocysts decreases the number and virulence of transmissible sporozoites[19,20]. As sporogony occurs after completion of the mosquito reproductive cycle, the oocysts do not directly compete with the vector but rely on unconsumed resources stored in the mosquito tissues[19,21].

Hormonal signals, in particular 20E, orchestrate the transcriptional and post-transcriptional networks that shape the mosquito post-blood meal metabolism[1,2,7]. Indeed, 20E-driven metabolic changes are crucial for release of stored and blood meal-derived nutrients for mosquito reproduction as well as parasite development[21]. MicroRNAs contribute to post-transcriptional regulation and link the endocrine regulation with metabolic homeostasis in *Aedes* mosquitoes[22–24]. Using a transcriptomic approach in *A. gambiae*, we identified three miRNAs whose fat body expression was induced shortly after blood feeding, namely miR-275,

miR-276, and miR-305[25]. Consistently with our findings, miR-275 was reported to regulate blood meal digestion and egg development[24], while miR-305 was shown to impact mosquito microbiota and *P. falciparum* development by an as yet unknown mechanism[26]. Here, we report the role of miR-276 in fine-tuning the expression of *branched chain amino acid transferase* (*BCAT*) in the fat body. We show that miR-276 depletion prolongs high levels of *BCAT* expression and AA catabolism in the fat body, thereby increasing female fertility. Unexpectedly, sustained high AA catabolism compromises *P. falciparum* sporogonic development and reduces the number of transmissible sporozoites. Our results demonstrate the important role of mosquito metabolism in vector competence and malaria transmission.

## Results

**AAs and ecdysone regulate miR-276 in the fat body.** To define the role of miR-276 in the mosquito reproductive cycle, we examined its expressional profile by reverse transcription quantitative real-time PCR (RT-qPCR) in the abdominal carcass tissues collected before and after blood feeding. As the fat body is highly enriched in abdominal carcasses, we refer to them as fat body samples hereafter. Newly-eclosed females showed high levels of miR-276 expression that declined on the second day after eclosion (Supplementary Fig. 1). A blood meal transiently increased miR-276 levels from 28 to 44 hpb (Fig. 1a). This expression pattern paralleled the kinetics of the steroid hormone ecdysone, a key regulator of mosquito development and reproduction, whose synthesis is triggered by a blood meal. To compare the timing of miR-276 expression and ecdysone titers, we measured the levels of the metabolically active form of ecdysone, 20-hydroxyecdysone (20E), in the blood-fed females by an enzyme-linked immunosorbent assay (ELISA) (Fig. 1a). 20E titers preceded the induction of miR-276 expression as they increased from 6 to 24 hpb and fell down to basal levels at 36 hpb. Based on these results, we hypothesized that expression of miR-276 in the fat body may be regulated by 20E.

As both 20E and AAs regulate the mosquito reproductive cycle, we gauged expression levels of miR-276 after stimulation with 20E or/and AAs in an ex vivo fat body culture system[27,28]. The abdominal carcasses were incubated in culture medium supplemented or not with 20E or/and AAs in vitro. Addition of AAs or 20E increased miR-276 levels by two-fold. Moreover, a significant five-fold induction of miR-276 expression was observed when AA and 20E were added together (Fig. 1b). These results suggested a synergistic effect of AA and 20E on regulation of miR-276 expression in the fat body after a blood meal. In line with these results, bioinformatics analysis identified one potential binding site for the ecdysone-induced transcriptional factor Broad in the upstream regulatory region of the miR-276 gene.

**miR-276 inhibits the branched chain amino acid transferase.** MicroRNAs post-transcriptionally promote transcript degradation or inhibition of translation by binding to specific sites located predominantly in the 3′UTR. To identify putative miR-276 mRNA targets, we applied three in silico target prediction tools, namely MiRanda[29], RNAhybrid[30] and MicroTar[31](Supplementary Data 1). We next used a publicly available RNAseq dataset[7] to identify the fat body-expressed candidate target mRNAs with the reversed expression pattern to miR-276. This dual approach allowed us to down-select eight putative targets for further analyses: *ATP synthase* (AGAP012081), *branched chain amino acid transferase* (*BCAT*; AGAP000011), *glycerol-3-phosphate dehydrogenase* (*GPD*; AGAP004437), *methylmalonate-semialdehyde dehydrogenase* (*MMSDH*; AGAP002499), *NADH dehydrogenase* (*NADH (I)*, AGAP010464), *3-methylcrotonyl-CoA*

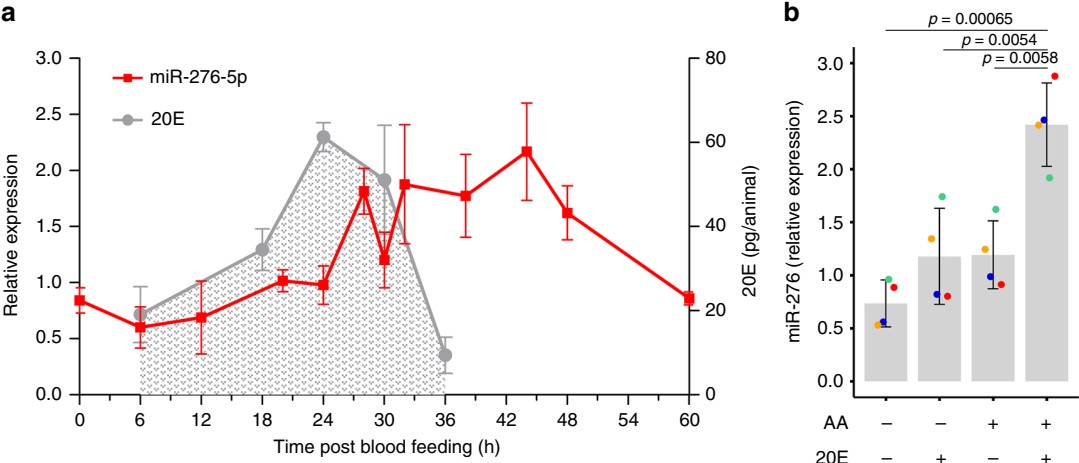

**Fig. 1 Amino acid (AA) and steroid hormone (20E) signalling regulate miR-276 expression. a** miR-276 (red) expression in the female fat body ($n = 10$, $N = 6$) and 20-hydroxyecdysone (20E) titres (gray) in the mosquito females ($n = 9$, $N = 3$) at different time points after blood feeding. The plots show mean ± SEM of independent experiments. **b** miR-276 expression in ex vivo fat body cultures of 3–4-day-old females ($n = 3$, $N = 4$) after incubation with medium with or without amino acids (AA) and 20E. miRNA expression levels were normalized using the ribosomal protein *RPS7* gene. Boxplots show the median with first and third quartile, whiskers depict the min and max of independent experiments, colored dots show mean of each experiment. The statistical significance was tested by one-way ANOVA followed by Tukey's post hoc test and the obtained *p*-values are shown above the horizontal lines ($n$ = number of mosquitoes pooled for each independent experiment; $N$ = number of independent experiments).

*carboxylase* (*MCC*; AGAP010228), *branched-chain alpha-keto acid dehydrogenase* (*BCKDH*, AGAP003136) and *mitochondrial ornithine receptor* (*MOR*; AGAP000448).

We postulated that expression of these target transcripts should be repressed by miR-276 and measured transcript levels of the selected mRNAs in the fat bodies of miR-276 knockdown mosquitoes (miR-276[KD]) at the peak of miR-276 expression (38 hpb) by RT-qPCR. miR-276 was silenced by injection of antisense oligonucleotides (miR-276 antagomirs), whereas injection of a scrambled antagomir served as control (Supplementary Table 2). Out of the examined candidates, only expression of *BCAT* significantly increased after miR-276 inhibition (Fig. 2a). *BCAT* carries one predicted miR-276 binding site in its 3′UTR (Fig. 2b). However, as we did not perform genome-wide analysis of mRNA changes following miR-276 inhibition, we cannot exclude that miR-276 regulates other mRNAs than *BCAT*.

We next tested for direct interaction between miR-276 and the binding site in the *BCAT* 3′UTR using a Dual Luciferase Reporter Assay in the *Drosophila* Schneider 2 (S2) cell cultures in vitro. This assay examines the ability of miR-276 to post-transcriptionally inhibit a reporter construct containing the *Renilla* luciferase reporter gene fused to the *BCAT* 3′UTR. We generated three constructs: (1) a positive control, which carries three copies of the miR-276 binding site (3x binding site), (2) the wild-type *BCAT* 3′UTR with an intact single miR-276 binding site (BCAT 3′UTR); and (3) a *BCAT* 3′UTR negative control with the mutated sequence of miR-276 binding site (BCAT 3′UTR[mut]). S2 cells expressed low levels of endogenous *Drosophila* miR-276 leading to basal inhibition of the *BCAT* 3′UTR reporter activity. Co-transfection of the mosquito miR-276 further enhanced inhibition of the luciferase reporter fused to the functional 3′UTR binding sites but not to the mutated control (Supplementary Fig. 2). Importantly, mosquito miR-276 significantly repressed expression of the positive control reporter with the three copies of the miRNA binding site and the wild-type *BCAT* 3′UTR reporter. In contrast, mutations in the BCAT miRNA binding site abolished the inhibition of the luciferase reporter by more than two-fold (Fig. 2c and Supplementary Fig. 2). These results demonstrated that *BCAT* 3′UTR is a *bona fide* target of miR-276.

To confirm our in vitro findings, we examined *BCAT* transcript levels in vivo after blood feeding in miR-276[KD] and scramble-injected mosquitoes. We observed that in the presence of miR-276, *BCAT* transcript levels went down as early as 24 hpb, inhibition of the miRNA function by antagomir prolonged *BCAT* expression by almost 12 h (Fig. 2d). We concluded that miR-276 tunes down *BCAT* expression during the late phase of the mosquito reproductive cycle.

**miR-276 fine-tunes AA metabolism of the reproductive cycle.** BCAT catalyzes the first step of the branched chain amino acid (leucine, valine, and isoleucine) catabolism. As miR-276 post-transcriptionally repressed *BCAT* during the late phase of the reproductive cycle, we examined the amino acid metabolism after miR-276 knockdown. We compared metabolite profiles of miR-276[KD] and control females (whole mosquitoes) at three time points after *P. falciparum*-infected blood feeding (10, 38, and 48 hpb) using gas chromatography-mass spectrometry (GC-MS) and liquid chromatography-mass spectrometry (LC-MS) (Supplementary Data 1). We selected time points before (10 hpb) and during miR-276 expression (38 and 48 hpb). In total, we detected 4,716 chromatographic peaks (*m/z* at a specific retention time) of which 816 were annotated in reference of authentic standards. The annotated data were normalized to the sample median and filtered using the interquartile range. The data were further log₂-transformed and pareto scaled for heatmap visualization. 294 features significantly changed across the reproductive cycle (two-way ANOVA; Supplementary Data 2) and were included in the heatmap. Massive metabolic changes after blood feeding were observed in both control and miR-276[KD] mosquitoes (Fig. 3a). As early as 10 hpb, a large cluster of highly enriched metabolites correlated with the influx of human blood (Cluster I). This cluster comprised amino acids, short and long chain fatty acids, as well as sphingomyelins, potentially representing metabolites derived from the ingested blood. A second cluster (Cluster II), comprising predominantly dipeptides, was abundant at 10 and 38 hpb and disappeared at the late phase of the reproductive cycle (48 hpb). Conversely, Cluster III featured the metabolites whose levels increased at 38 and 48 hpb, corresponding to the periods of

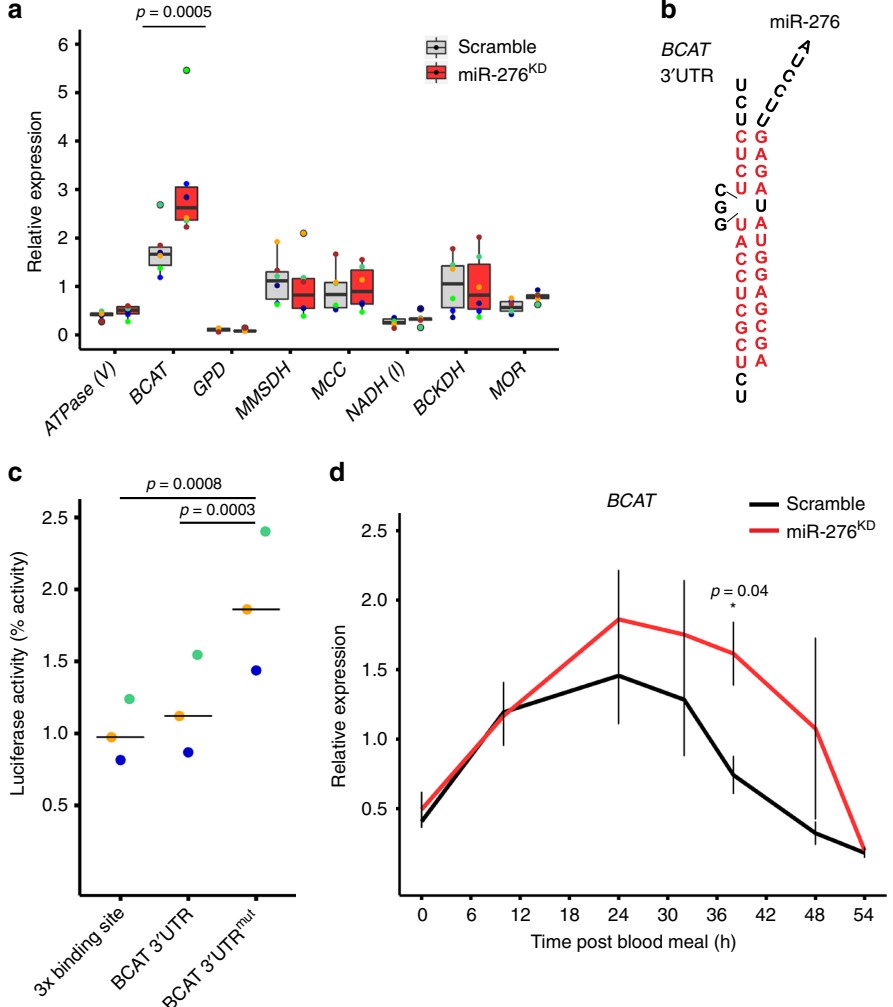

**Fig. 2 miR-276 post-transcriptionally represses the *branched-chain amino acid transferase* in the mosquito fat body. a** Expression of predicted miRNA targets at 38 h post blood feeding in the fat body of mosquitoes ($n = 5$, $N = 6$) injected with anti-miR-276 (miR-276, red) or scrambled antagomir (scramble, gray). Expression of the *ATP synthase* (ATPase V), *branched chain amino acid transferase* (BCAT), *glycerol-3-phosphate dehydrogenase* (GPD), *methylmalonate-semialdehyde dehydrogenase* (MMSDH), *3-methylcrotonyl-CoA carboxylase* (MCC), *NADH dehydrogenase* (NADH (I)), *branched-chain alpha-keto acid dehydrogenase* (BCKDH) and *mitochondrial ornithine receptor* (MOR). Expression levels were normalized using the ribosomal protein *RPS7* gene. Boxplots show the median with first and third quartile, whiskers depict the min and max. Statistical significance was tested by two-way ANOVA followed by Tukey's *post hoc* test and significant differences are shown by the *p*-value above the horizontal line. **b** Sequence of the predicted miR-276 binding site in the *BCAT* 3′UTR. Red color indicates complementary nucleotides in the 3′UTR binding site and miR-276. **c** Dual luciferase reporter assay in vitro. A construct containing three copies of miR-276 binding sites served as positive control. Lines depict medians and colored dots show means of independent experiments ($N = 3$). Statistical significance of differences was tested by one-way ANOVA followed by Tukey's post hoc test and significant differences are indicated by the *p*-values above the horizontal lines. **d** *BCAT* expression in the fat body during the reproductive cycle of mosquitoes injected with anti-miR-276 (miR-276KD, red) or scrambled antagomir (scramble, black). Expression levels were normalized using the ribosomal protein *RPS7* gene. Means ± SEM (vertical lines) were plotted. Statistically significant differences between miR-276KD and control mosquitoes were determined by two-way ANOVA followed by Tukey's post hoc test ($N = 3$) and significant differences are shown by the *p*-value (*) ($n$ = number of mosquitoes pooled for each independent experiment; $N$ = number of independent experiments).

complete blood digestion and ovary development. The vast majority of Cluster III metabolites were lipids (triacylglcerides (TAGs), phosphatidiylcholines, phosphatidiylethanolamines, phosphatidiylinositols), and trehalose (Fig. 3a). Overall, the observed metabolite dynamics were not significantly perturbed by miR-276KD and closely recapitulated the major physiological processes induced by blood feeding. This observation was in line with the fine-tuning role of miR-276, whose silencing did not hinder the overall blood feeding-induced metabolic profile.

We next focused on BCAT-related metabolites. BCAT initiates BCAA catabolism by transferring the BCAA amino group to α-ketoglutarate forming glutamate, which is thought to serve as a nitrogen sink in insects[32] (Fig. 3b). While BCAA levels did not massively change in miR-276KD mosquitoes, significantly higher levels of glutamate were detected at 48 hpb (Fig. 3c). Furthermore, miR-276KD also increased levels of histidine, lysine, citrulline, arginine, argininosuccinate and ornithine at 48 hpb (Fig. 3d), pointing to an accumulation of metabolites associated with nitrogen metabolism, the waste product of AA catabolism, during the late phase of reproductive cycle. The detected increase in nitrogen metabolism in miR-276KD mosquitoes during the late phase of the reproductive cycle (48 hpb) is consistent with the transcriptional increase in *BCAT* levels observed in miR-276KD females. Taking together, our results suggest that miR-276

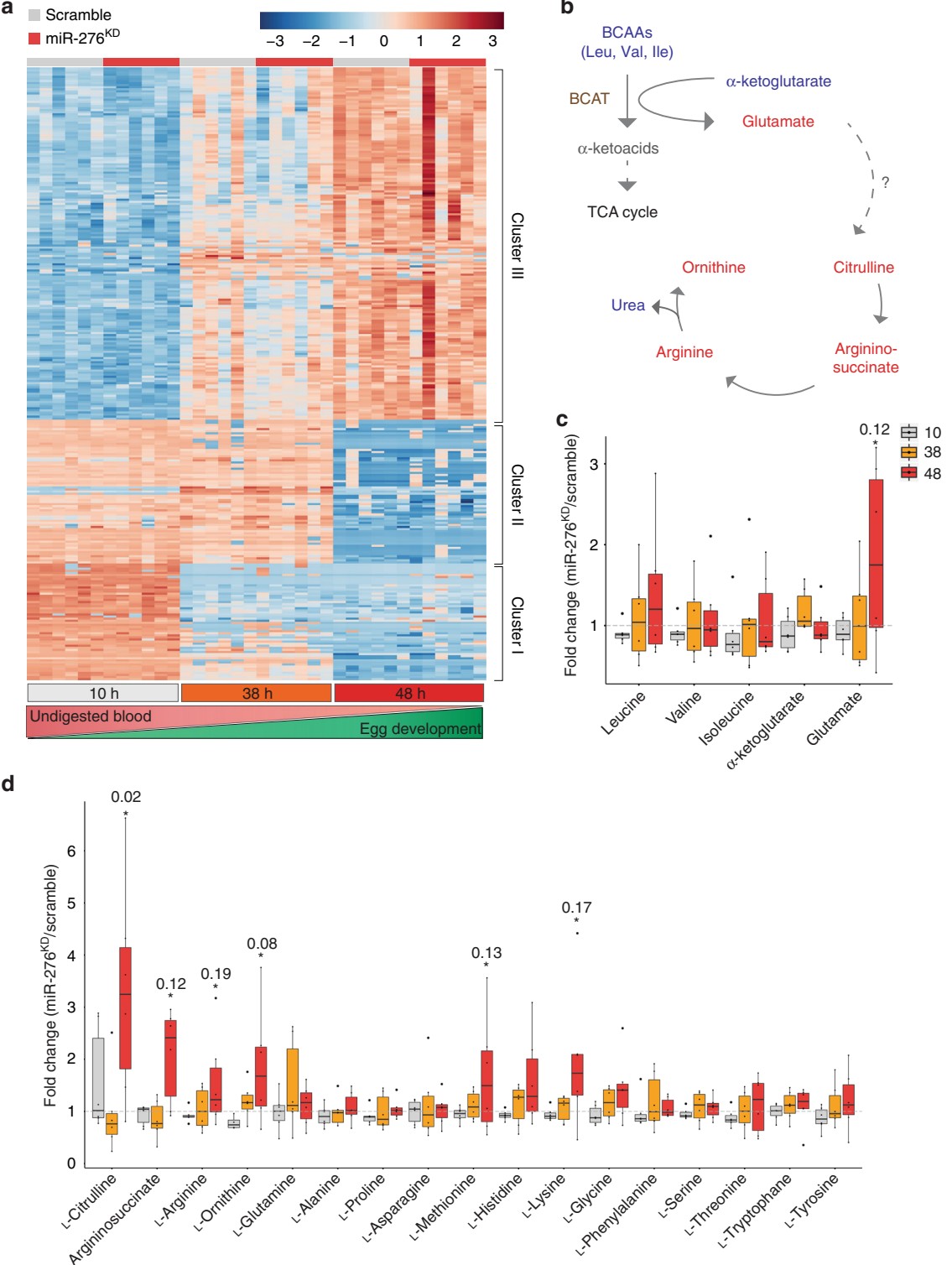

contributes to the termination of the AA catabolism by fine-tuning *BCAT* transcript levels and that miR-276 silencing prolongs the catabolic phase of the mosquito reproductive cycle.

**miR-276 regulates oogenesis and *P. falciparum* sporogony.** Both mosquitoes and malaria parasites rely on nutrients for successful ovary and sporogonic development, respectively. We first examined how prolongation of the catabolic phase in miR-276^KD mosquitoes affects oogenesis. Female mosquitoes

were injected with the miR-276 or control (scramble) and blood fed three days later to induce egg development. Two days later, females were transferred into egg laying chambers and the numbers of eggs laid by individual females were enumerated. Inhibition of miR-276 significantly increased egg laying (fertility) without compromising larval hatching rates (fecundity) (Fig. 4a and Supplementary Fig. 5). To provide further evidence that *BCAT* is the target of miR-276, we examined the effect of *BCAT* knockdown on oogenesis. In line with the results obtained for

**Fig. 3 miR-276 fine-tunes amino acid (AA) metabolism in the late phase of the mosquito reproductive cycle. a** Heatmap of annotated metabolite levels that vary across the reproductive cycle. Metabolite levels were measured by GC-MS and LC-MS from whole females ($n = 10$, $N = 6$) at 10, 38, and 48 h post *P. falciparum*-infected blood meal in control (scrambled, gray) and miR-276-depleted (miR-276KD, red) mosquitoes. Bottom bars indicate time points after blood feeding (10 h, gray; 38 h, orange; 48 h, red), and gradients below show the progression of blood digestion (red) and ovary development (green). Samples are plotted on the *x*-axis and metabolites on the *y*-axis. For heatmap visualization, data were log$_2$ transformed and pareto scaled. **b** Overview of the classical enzymatic activity of the branched chain aminotransferase (BCAT) in animals. The amino group of BCAAs is transferred to α-ketoglutarate by BCAT resulting in glutamate. Glutamate serves as a nitrogen sink, which is passed on to the urea cycle in vertebrates. As mosquitoes lack the carbamoyl phosphate synthase, the link between glutamate and citrulline of the urea cycle is unknown. Within the classical urea cycle, citrulline is converted to argininosuccinate and subsequently arginine, which is metabolized to ornithine ultimately releasing urea. Metabolite changes upon miR-276 inhibition (48 hpb) are shown in red, unchanged metabolites are in blue, and undetected metabolites are in gray. **c** Fold-change differences in branched chain amino acids (leucine, valine, isoleucine), α-ketoglutarate and glutamate detected in miR-276KD as compared to scramble controls. **d** Fold change differences in metabolites of AA metabolism at 10 h (gray), 38 h (orange), 48 h (red) post blood feeding in whole females. Statistical significance of differences in metabolite levels of miR-276KD and scramble controls was tested by *t*-test ($n$ = number of mosquitoes pooled for each independent experiment; $N$ = number of independent experiments).

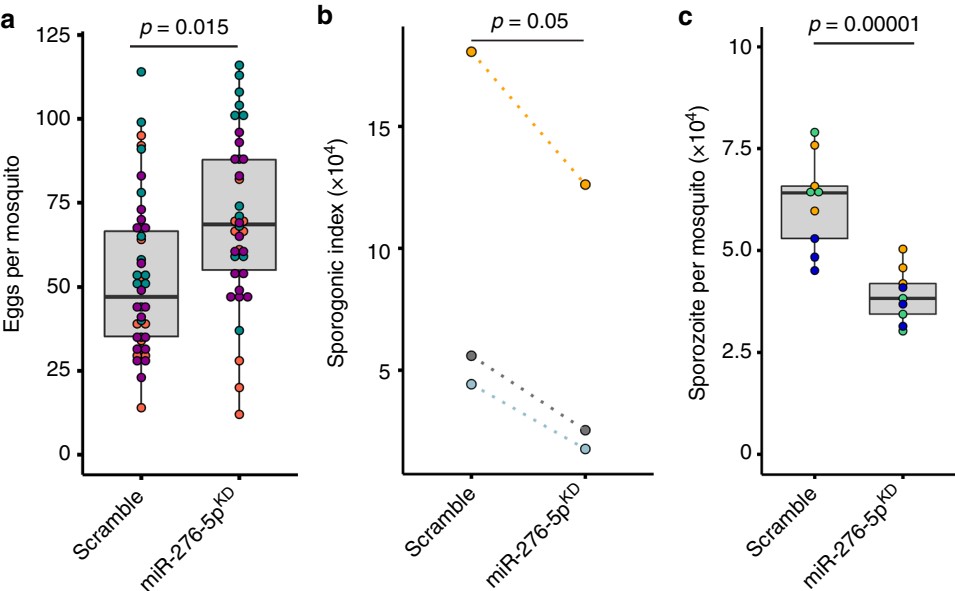

**Fig. 4 Effects of miR-276 silencing on mosquito fertility and *P. falciparum* sporogonic development. a** Egg laying rates (fertility) of individual females injected with miR-276 (miR-276KD) or control antagomir (scramble) after a blood meal. Boxplots show the median with first and third quartile, whiskers depict min and max values. Each dot represents one mosquito and dot colors indicate independent experiments ($N = 3$, Supplementary Table 3). **b** Females injected with miR-276 (miR-276KD) or control (scramble) antagomir were infected with *P. falciparum*. At day 11 post infection, oocyst number and size were measured. The median of *P. falciparum* oocysts per midgut and mean of *P. falciparum* oocyst size per experiment were multiplied to generate a sporogonic index ($N = 3$, Supplementary Table 3). **c** The numbers of the salivary gland sporozoites per mosquito were quantified on day 14 post infection. Boxplots show the median with first and third quartile, whiskers depict min and max values. Colors show replicates of each independent experiment. Statistically significant differences are indicated by *p*-values above the horizontal lines deduced by one-way ANOVA and Tukey's post-hoc test ($N = 3$, Supplementary Table 3), $N$ = number of independent experiments.

miR-276KD, *BCAT* silencing by injection of dsRNA prior to blood feeding decreased egg laying, suggesting a critical role of amino acid catabolism in mosquito reproduction (Supplementary fig. 6). We concluded that miR-276-BCAT axis regulates mosquito metabolic investment into oogenesis by terminating the catabolic phase before complete nutrient consumption by oogenesis.

We next asked whether the observed shift in the metabolic investment into oogenesis impacts parasite development. Using the same experimental settings, we infected miR-276KD and control mosquitoes with *P. falciparum* and gauged the number and size of oocysts 11 days post infection. Similarly to previous reports, wounding by injection of scramble antagomir decreased parasite infection load[33] (Supplementary Fig. 4). Importantly, inhibition of miR-276 further significantly reduced either the median number or the mean size of *P. falciparum* oocysts. To quantify oocyst

development, we calculated a sporogonic index by multiplying the median number by the mean size of oocysts per midgut in each independent experiment. We observed that regardless of infection levels, the sporogonic index was significantly lower in miR-276KD mosquitoes than in controls (Fig. 4b). We further examined whether the observed decrease in the parasite mass translated into lower numbers of the salivary gland sporozoites. Indeed, miR-276 inhibition significantly reduced by two-fold the loads of the salivary gland sporozoites (Fig. 4c), suggesting that prolongation of the catabolic phase restricts resources available for the development of the parasite transmissible forms.

In summary, our results demonstrated that the miR-276-regulated switch from catabolic to anabolic amino acid metabolism in the fat body restricts mosquito investment into oogenesis and benefits *P. falciparum* development.

## Discussion

The hematophagous lifestyle endows mosquitoes with a speedy within-days egg development, but also benefits the transmission of the malaria parasite. Blood feeding induces massive metabolic changes that re-direct all resources towards reproduction, a state that is unsustainable over a long period of time. Here, we provide evidence that miR-276 contributes to a negative feedback loop that resets mosquito metabolism by repressing BCAA catabolism. This metabolic reset restricts mosquito reproductive investment and, thereby, benefits *P. falciparum* sporogony. Our results highlight a crucial role of mosquito reproduction in vector competence.

The fat body is the major insect storage organ, which produces the majority of hemolymph-born vitellogenic proteins and critically contributes to mosquito ovary development[34]. After blood feeding, high titers of AAs and 20E promote a metabolic switch in the fat body from anabolism to catabolism, resulting in a rapid release of glycogen and lipid reserves[1,2]. The same signals induce expression of negative regulators, including miR-276, that warrant the timely reset of the fat body metabolism[14,15,25]. Massive degradation of AAs in the fat body supports high energy demands of nutrient mobilization and protein synthesis for successful egg development[8,10]. However, high levels of AA degradation also generate a considerable amount of nitrogen waste, which may restrain the metabolic investment during the reproductive cycle[35–37]. Interestingly, *Aedes* mosquitoes lack the full repertoire of the urea cycle enzymes and rely on an alternative ammonia detoxification pathway engaging glutamine, alanine, and proline[32,38]. We did not observe miR-276-driven changes in these metabolites at any tested time point. Instead, we detected increased levels of glutamate and other classic urea cycle metabolites (citrulline, ornithine, argininosuccinate, and arginine) during the late phase of the reproductive cycle. These results point to potentially important metabolic differences between *Aedes* and *Anopheles* mosquitoes and call for further targeted metabolome investigations using $N^{15}$-labelled BCAAs to determine the urea cycle components in both mosquito species. The timing and nature of metabolic changes observed in this study suggest that miR-276 contributes to fine regulation of AA catabolism in the late phase of the reproductive cycle. During this phase, the fat body undergoes a reverse shift from catabolic to anabolic metabolism in order to replenish nutrient resources in preparation for the next reproductive cycle[1,2]. As part of this metabolic switch, miR-276 post-transcriptionally represses

translation of the mRNA encoding the branched chain amino acid transferase, the catalyzer of the first step of BCAA breakdown. BCAA turnover represents a central hub that shapes the overall metabolic state and is tightly regulated[39–42]. In *Drosophila*, BCAA degradation is controlled by miR-277[43]. Interestingly, the BCAA degradation pathway in *Aedes* shapes the bacterial load in the midgut by an as yet unknown mechanism[44]. Therefore, in addition to the fat body examined here, post-transcriptional regulation of BCAA metabolism by miR-276 may affect mosquito physiology at multiple levels, including the midgut microbiome. Our results show that inhibition of miR-276 prolongs high levels of BCAA catabolism and benefits female fertility. These conclusions are further supported by the observed changes in expression of the major yolk protein gene *Vitellogenin* (*Vg*) (Supplementary Fig. 3). As *Vg* lacks miR-276 binding sites in its 3′UTR and cannot be directly regulated by this miRNA, the prolongation of the AA catabolism must be coordinated with lipid metabolism, an interesting observation which calls for further investigation. Based on these results, we propose that miR-276 contributes to a negative feedback loop that shifts the catabolic metabolism induced by AAs and 20E during the early-/mid-phase of the female reproductive cycle to the anabolic phase. Timely attenuation of the catabolic phase is important to restrict the costly reproductive investment and prevent complete nutrient exhaustion (Fig. 5a). Metabolic costs likely play an important role in the regulation of reproductive investment as blood digestion in hematophagous insects escalates oxidative stress and nitrogen waste products[36,37,45]. Indeed, blood digestion reduces the mitochondrial activity of the flight muscles, presumably to mitigate the overall metabolic stress[46]. Furthermore, timely conclusion of the reproductive program is essential in preparation for the next reproductive cycle[14,15]. We propose that the duration of the reproductive cycle is tightly regulated by a balance between resource availability and reproductive costs.

Malaria parasites critically rely on mosquito hematophagy for sexual reproduction and transmission. Earlier studies focused on the trade-off between mosquito reproduction and *Plasmodium* development comparing reproductive readouts of infected versus uninfected mosquitoes[47,48]. However, mosquito females complete egg development before massive parasite proliferation, thereby avoiding direct competition for nutrients with *Plasmodium* parasites[19,21]. Here, we experimentally demonstrate that prolonging high levels of the AA catabolic phase by inhibition of miR-276 increases mosquito reproductive investment and

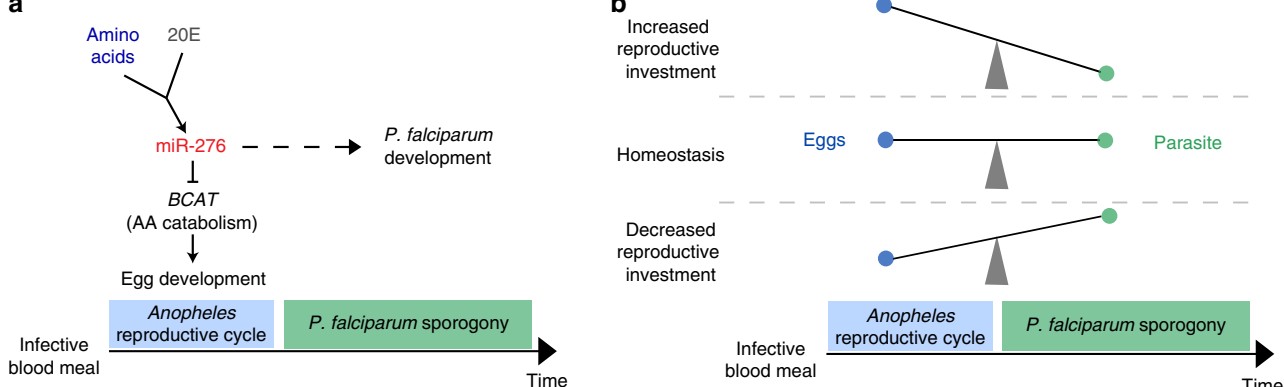

**Fig. 5 Summary of miR-276 function and model of mosquito reproductive investment and *P. falciparum* development interaction. a** Amino acids (AA) and 20-hydroxyecdysone (20E) induce expression of miR-276 after blood feeding and, thereby, initiate the termination of the fat body AA catabolism by inhibiting the *branched chain amino acid transferase* (*BCAT*). This catabolism to anabolism switch in the fat body restricts mosquito investment into reproduction. *P. falciparum* sporogonic stages parasitize spare mosquito resources for their own development. **b** Mosquito nutritional status and the extent of the reproductive investment shape the within-vector environment which shapes *P. falciparum* sporogonic development.

restricts the sporogonic development of *P. falciparum*. We propose that the balance between the mosquito metabolic status and reproductive investment determines *Plasmodium* sporogony and transmission (Fig. 5b). Importantly, variability in reproductive investment strategies could be one of the factors underlying the differences in the vector competence observed between mosquito species that are equally exposed to *Plasmodium* infections[49].

Together with the previous reports[19,21], our study describes a general mechanism by which genetic or hormonal manipulations of the mosquito metabolic program and resource allocation impinge on female fertility, within-vector *Plasmodium* development and within-host virulence. We propose that the mosquito metabolic status and reproductive investment are essential components of vector competence that shape malaria transmission.

## Methods

**Mosquitoes**. *Anopheles coluzzii* Ngousso (*TEP1*S1*) strain was used throughout the study. Mosquitoes were maintained at 30 °C 80% humidity at 12/12 h day/night cycle with a half-hour long dawn/dusk period. All mosquitoes were fed *ad libitum* with 10% sugar solution.

**Blood meal and *P. falciparum* infections**. Asexual cultures (parasitaemia >2%) of *P. falciparum* NF54 (kindly provided by Prof. R. Sauerwein, RUMC, The Netherlands) were harvested by centrifugation for 5 min at 1,500 rpm and diluted with fresh red blood cells (RBS) (Haema) to 1% total parasitaemia at 4% hematocrit. Asexual and gametocyte cultures were incubated at 37 °C with 3% $O_2$ and 4% $CO_2$. Gametocyte medium was changed daily for 15–16 days on heated plates to reduce temperature drop. On day 14 after seeding, gametocytaemia was gauged by Giemsa-stained smears, and parasite exflagellation rates were evaluated by microscopy.

Mosquitoes were fed using an artificial feeder system[50,51] with diluted gametocyte cultures 15–16 days after seeding at a final gametocytaemia of 0.15–0.22%. The feeding system was prepared by covering the bottom of the feeders with a stretched parafilm. The feeder was heated to 37 °C by water flow. Mosquitoes were fed for 15 min and unfed mosquitoes were removed. Only fully engorged females were kept for further analyses at 26 °C 80% humidity.

**Parasite quantification**. For *P. falciparum* oocyst counts, infected mosquitoes were killed in 70% ethanol on day 11 after infection. Mosquitoes were washed twice in PBS, midguts were dissected in 1% mercurochrome in water and incubated for 10 min at room temperature. Oocysts were counted under a stereoscope. For oocyst size, three representative images of midgut oocysts were taken per midgut by light microscope at 40x magnification and oocysts size was gauged by Fiji Image. The sporogonic index was calculated by multiplying the median number of oocysts and the mean oocyst size per experiment.

The numbers of *P. falciparum* sporozoites in the salivary glands were evaluated 14 days post infection. At least 30 mosquitoes per condition were killed in 70% ethanol, washed with PBS and the salivary glands were dissected in RPMI medium (Gibco) 3% bovine serum albumin (Sigma Aldrich). The salivary glands were homogenized with a glass plunger to release the sporozoites. The homogenate was filtered twice with cell strainers (100 and 40 μm) into glass vials and diluted at 1:10, 1:20, and 1:50. The sporozoites were counted using a hemocytometer under a light microscope. Sample sizes per experiment are provided in Supplementary Table 3.

**20-hydroxyecdysone (20E) quantification**. 20E levels were quantified using the 20E EIA kit (Cayman Chemical). Nine females per time point were collected into the microtubes, snap frozen and kept at −80 °C. On the day of analysis, all samples were homogenized in 500 μl methanol using steal beads and a Qiagen tissue lyser at 50 rpm for 10 min. Samples were centrifuged at 15,000 rpm for 1 min and the supernatant (420 μl) was transferred to a clean tube. The methanol was evaporated in a speedvac. The pellets were dissolved in 230 μl of EIA buffer (Cayman Chemical). A 1:3 dilution series of 20E starting from 361 ng/ml served as a standard curve. Measurements were performed according to the manufacturer's instructions.

**Ex vivo fat body tissue culture**. Ex vivo fat body culture was performed as following: amino acid, salt, calcium and TRIS buffer stock solution was prepared according to Chung et al. [27], sterile filtered and stored at −20 °C. The basic medium was prepared without amino acids, which were replaced with an equal molarity of mannitol. 20E was dissolved in ethanol and equal amounts of ethanol were added to the medium control. Three abdominal carcasses of 3–4-day-old females were dissected and incubated with: (i) medium, (ii) $10^{-6}$ M 20E, (iii) amino acids (AAs); and (iv) $10^{-6}$ M 20E (Sigma) and AAs[27]. The abdominal carcasses cultures were incubated for 6 h at 27 °C. RNA was isolated by RNAzol for quantification of miRNA/mRNA expression.

**Sample collection for RNA expression analysis**. For miRNA and mRNA expression kinetics, ten sugar-fed or blood-fed females were dissected on ice at different time points after a blood meal. The abdominal carcasses (the fat body) were immediately homogenized in RNAzol (Sigma Aldrich) and kept at −80 °C for RNA isolation. Total RNA was isolated by RNAzol according to the manufacturer's recommendations including the purification step with 4-Bromoanizol. The total RNA yield was measured with a Qubit (Thermo Fisher Scientific).

**miRNA expression analysis**. miRNA expression was quantified using the miScript PCR System (Qiagen) that allows real-time PCR quantification of mRNA and miRNA expression from the same reverse transcription reaction mix. The cDNA levels were further measured by Quantitect SYBR Green PCR kit (Qiagen). Forward miR-276 primer was obtained from the miScript primer assay and reverse primer was synthesized using the sequence stated in the Supplementary Table 1[24]. Expression data were calculated by the relative standard curve method. Relative quantities of miRNA expression were normalized to the gene encoding the ribosomal protein S7 (*RPS7*).

**mRNA expression analysis**. For target identification, RNAs were reverse transcribed using the RevertAid H Minus First Strand cDNA synthesis kit (Thermo Fisher Scientific) and analyzed by RT-qPCR using the SYBR Green PCR mix (Thermo Fisher Scientific). Expression data was calculated using relative standard curve method. Relative quantities of miRNA expression were normalized to the gene encoding ribosomal protein S7 (*RPS7*). Primers sequences are listed in Supplementary Table 1.

**Computational miRNA target prediction**. miRNA targets were predicted in silico using three independent algorithms: miRANDA[29], RNAhybrid[30], and microTAR[31].

**miRNA inhibition**. Antagomirs (anti-miR-276-5p and a scrambled version of the same antagomir that has no target in *A. gambiae* genome) were designed using the RNA module for custom single-stranded RNA synthesis (Dharmacon) as RNA antisense oligos with 2′-O-methylated bases, a phosphorothioate backbone at the first two and last four nucleotides and a 3′cholesterol (Supplementary Table 2). For miRNA inhibition, mosquitoes were anesthetized with $CO_2$ and microinjected with 207 nl of 200 μM antagomir (41 pmol/female) at 12–18 h post eclosion using the Drummond NanoJect II (Drummond Scientific). Mosquitoes were left for four days to recover before a blood feeding or *P. falciparum* infection.

**Egg laying and hatching assay**. Female and male mosquitoes were maintained in the same cage. For fertility assays, individual females were gently transferred into single cups with egg dishes (wet Whatman paper) on day three after blood feeding and allowed to oviposit for two nights. Eggs were counted by stereoscope. For fecundity assays, females were kept together after a blood feeding and allowed to lay eggs on a common egg dish. After egg laying, the egg dish was kept in water for one day at 26 °C 80% humidity. Larval hatching rates were gauged by counting the number of open egg shells (at least 100 eggs) using a stereoscope.

**Dual luciferase assay**. In vitro target validation was performed using *Drosophila* S2 cells (Invitrogen). The cells were kept in Schneider *Drosophila* medium supplemented with 10% heat-inactivated FBS (Gibco) and 1% Penicillin Streptomycin (vol/vol) (Thermo Fisher Scientific) at 25 °C. Three luciferase constructs were examined: (1) a positive control containing three sites reverse complementary to miR-276 with four nucleotide linkers between each miR-276 binding site; (2) *A. gambiae BCAT* 3′UTR and (3) *A. gambiae BCAT* 3′UTR with a scrambled miR-276 binding site. All constructs were separately inserted into the multiple cloning region located downstream of the *Renilla* translational stop codon within the psiCheck-2 vector (Promega). psiCheck-2 reporters (100 ng) and a synthetic *A. gambiae* miR-276-5p miScript miRNA Mimic (100 ng, Qiagen) at a final concentration of 100 nm were co-transfected into *Drosophila* S2 cells using FuGENE HD transfection reagent (Promega). Cells transfected only with psiCheck-2 reporters were used as a "no miRNA mimic" control. Dual Luciferase Reporter Assay was performed 48 h post transfection using the Dual Luciferase Reporter Assay System (Promega). Firefly luciferase in the psiCheck-2 Vector was used for normalization of the *Renilla* luciferase expression. Measurements were made in triplicates, and transfections were repeated three times independently.

**Sample preparation for untargeted metabolomics**. Female mosquitoes (1-day-old) were injected with anti-miR-276-5p or scrambled antagomirs as described above. Four days later, mosquitoes were infected with *P. falciparum*. At 10, 38, and 48 hpb, ten females per treatment were snap-frozen in liquid nitrogen. The samples were stored at −80 °C until preparation. Additionally, at 38 and 48 hpb, abdominal carcasses (the fat body) of ten female mosquitoes per group were collected for miRNA target analysis by RT-qPCR.

Untargeted metabolomics was performed by the company MetaSysX GmbH, Potsdam, Germany. The sample preparation was performed according to MetaSysX standard procedure, a modified protocol from Salem et al.[52]. Whole

bodies of ten snap-frozen mosquitoes were grounded into a homogenous fine powder using a tissue homogenizer. The polar metabolites and lipophilic compounds were extracted by methyl-tert-butyl-ether (MTBE)/methanol/water solvent system that separates molecules into aqueous and organic phase, respectively. After MTBE extraction, the organic phase containing lipids and lipophilic compounds were transferred to 1.5 ml tube. The leftover of the organic phase was removed with a vacuum aspirator and 500 μl of aqueous phase containing semi-polar and polar compounds was collected to a new tube. Additionally, 150 μl of the polar phase were transferred to a fresh tube for subsequent derivatization and GC-MS analysis. All samples were dried down using a centrifugal evaporator and stored at −80 °C until LC-MS and GC-MS analysis.

**Liquid chromatography mass spectrometry**. For polar and lipid measurements, dried samples were resuspended in 170 μl of water or acetonitrile, respectively. A 2 μl injection volume was analyzed. Small molecules were separated by ultra-performance liquid chromatography (UPLC) and analyzed on an QExactive Orbitrap MS (Thermo Fisher Scientific) in positive and negative polarity.

The samples were measured with a Waters ACQUITY Reversed Phase Ultra Performance Liquid Chromatography (RP-UPLC) coupled to a Thermo-Fisher QExactive mass spectrometer. $C_8$ (100 mm × 2.1 mm × 1.7 μm particles; Waters) and HSS T3 $C_{18}$ (100 mm × 2.1 mm × 1.8 μm particles; Waters) columns were used for the lipophilic and the hydrophilic measurements, respectively. A 15 min gradient was used for separation of polar and lipophilic compounds. The mobile phases for separation of polar and semi-polar compounds were 0.1% formic acid in $H_2O$ (buffer A) and 0.1% formic acid in acetonitrile (buffer B). The chromatographic separation of these analytes was performed in the following conditions: A 99% initial to 1 min, A 99% to A 60% from 0 to 11 min, A 60% to A 30% from 11 to 13 min and A30% to A 1% from 13 to 15 min. The following mobile phases were used for lipids and lipophilic molecules separation: 1% of 1 M $NH_4Ac$ in 0.1% acetic acid (buffer A) and acetonitrile:isopropanol (7:3) containing 1% of 1 M $NH_4Ac$ in 0.1% acetic acid (buffer B). The separation of lipids and lipophilic compounds was performed with a step gradient from 45% initial to 1 min, 45 to 25% A in 4 min, 25 to 11% A in 11 min and 11 to 0% A in 15 min. Chromatograms were recorded in Full Scan MS mode (Mass Range [100-1,500]).

All mass spectra were acquired in positive and negative mode with the following settings of the instrument: Heated electrospray ionization (HESI) was used, spray voltage was 3.5 kV, capillary temperature −275 °C, sheath gas flow rate −60 units, mass resolving power 70,000, $3 × 10^6$ target value (AGC) and maximal fill time of 200 ms. For lipid annotation, all samples were pooled and measured four times in LC-MS/MS mode. The fragmentation was performed in data-dependent MS/MS mode using higher-energy collisional dissociation (HCD). Three most intense ions were selected for fragmentation per cycle using a normalized collision energy of 25. The full scan spectra were acquired in the 100-1,500 m/z range at a mass resolution of 35,000 with an AGC of $10^5$ ions with maximal fill time of 100 ms. The target values for data dependent MS/MS scans were set to $5 × 10^4$ ion with a maximal fill time of 50 ms and an isolation window of 1 Th. The MS/MS ions were measured at a resolution 17,500 and the dynamic exclusion was set to 3 s. Thermo Excalibur was used for the data acquisition.

An in-house metaSysX database of chemical compounds was used to match the features detected in the LC-MS polar and non-polar platform. The metaSysX database contains the mass-to-charge ratio and the retention time information of reference compounds measured at the same chromatographic and spectrometric condition as samples measurements. The matching criteria for the polar platform were 5 ppm and 0.095 min deviation from the reference compounds mass-to-charge-ratio and retention time respectively and for lipid annotation 5 ppm and the range (−0.095; 0.255 min) time deviation. Coeluting compounds with the same ion mass were kept as unsolved annotation. Lipid annotation was additionally confirmed by MS/MS generated fragments. A metaSysX-developed R-based algorithm uses information from fragments, neutral losses presence and precursor masses information detected in both ionisations' modes. This information is combined with the information obtained from the database search to give an additional level of confidence. The lipid database was created based on the precursor ion mass, fragmentation spectrum, and elution patterns[53]. The elution patterns of annotated lipids are used to provide an additional level of confidence for lipid annotation.

**Gas chromatography mass spectrometry**. Dried samples were suspended in methoxy-hydrochloride/pyridine solution and incubated at 30 °C for 90 min followed by derivatization with N-methyl-N-trimethylsilyltrifluoroacetamide (MSTFA) at 37 °C for 30 min[54]. The samples were measured on an Agilent Technologies GC coupled to a Leco Pegasus HT mass spectrometer which consists of an EI ionization source and a TOF mass analyzer. Column: 30 m DB35; starting temp: 85 °C for 2 min; gradient: 15 °C per min up to 360 °C.

NetCDF files that were exported from the Leco Pegasus software were imported into the "R" Bioconductor package TargetSearch[55] to transform retention time to retention index (RI), to align the chromatograms, to extract the peaks, and to annotate them by comparing the spectra and the RI to the Fiehn Library and to a user created library. Annotation of peaks was manually confirmed in Leco Pegasus. Analytes were quantified using a unique mass. Metabolites with a RT and a mass spectrum that did not have a match in the database were labelled as unknown.

**Data filtering and normalization**. The GC- and LC-MS datasets were normalized to the sample median intensity. For heatmap visualization, missing data were replaced with the smallest value of the detected metabolite. Furthermore, the remaining dataset was subjected to interquartile range filtering. The data was then log2 transformed and scaled by mean-centering and by division by the square root of each feature (pareto scaling). The raw GC-MS and LC-MS dataset is attached as Supplementary material (Supplementary Data 3).

**Reporting summary**. Further information on research design is available in the Nature Research Reporting Summary linked to this article.

## Data availability

The polar and lipid LC-MS data in positive and negative polarity as well as GC-MS data that support the findings of this study have been deposited in the MetaboLights repository with the accession code MTBLS1282. The paper is available as preprint on the bioRxiv server [https://doi.org/10.1101/548784]. Source data for Figs. 1, 2, 4 are provided in the Source Data file.

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

## Acknowledgements

The authors wish to thank Liane Spohr for mosquito breeding, Manuela Andres and Daniel Eyermann for *P. falciparum* cultures and Cornelia Kreschel for sporozoite isolation and counting. Finally, we are grateful to all members of the Vector Biology Unit for fruitful discussions and to Dr. Guilia Costa, Dr. Paola Carrillo-Bustamante, and Clare Newell for constructive comments on the manuscript.

## Author contributions

L.L. and E.A.L. conceived the study and designed the experiments. L.L. and M.J. performed experiments and analyzed the data. S.K. provided expertise on the untargeted metabolomics experiments. L.L. and E.A.L. wrote the paper.

## Competing interest

The authors declare no competing interest.
