## [Peer Review File · Nature Communications]

Reviewers' Comments:

Reviewer #1:

Remarks to the Author:

In this study, the authors investigated the blood meal-induced miR-276-5p in *Anopheles* mosquitoes and have shown that it affects amino acid catabolism by targeting the branched chain 93 amino acid transferase in the fat body. Antagomir depletion of this miRNA leads to the extension of the reproductive cycle, as evaluated by the egg number. Importantly, it is also affected *Plasmodium* development.

Overall, the findings of this work are novel and significant. However, I find it to be not thorough and perhaps rushed to a publication stage too prematurely. The analysis of miRNA-276 target prediction is good and satisfactory. However, in my opinion there are many shortcomings.

It is important to add wild type controls in most of physiological experiments and profile measuring, for example in Fig. 2D and 4. Utilization of the miR-276 scrambled probe is essential. It is, however, could be misleading because of a potential adverse effect of injection or potential effects of a foreign molecule introduction.

Measuring the effect of miR-276 on the postblood meal (PBM) phase should also be evaluated by such an important parameter as the duration of the Vg expression period and not only egg number, which does not show actual duration of this process. This will be a more accurate indicator of the delay in the cycle.

It is essential to utilize mimic phenotype rescue experiments. Will the reproductive cycle be restored following miR-276 antagomir and mimic applications; what is the mimic effect on the target enzyme expression; effect on the timing of vitellogenic period and egg number; finally on *Plasmodium* development.

Fig. 1 needs an explanation. The authors have shown that amino acids and 20-hydroxyecdysone (20E) work together in activation of miR-276 expression. Such regulation has been demonstrated for a number of mosquito genes that are expressed during the first half of the PBM phase (about 6-30h). However, miR-276 expression extends at its high level until 44 hr, way after the 20E titer declines to the background level (36 hr PBM). Thus, the regulation of the miR-276 expression is more complex than it has been presented here. To more thoroughly investigate miR-276 regulation, it would be advisable to investigate whether the effect of 20E on miR-276 is direct through its receptor EcR. Analysis of the miR-276 gene promoter for potential binding sites for EcR or its early gene E74 could be done here.

I find a part of study investigating the effect of miR-276 on *Plasmodium* development to be rather superficial. For example, one cannot exclude a direct effect of miR-276 on *Plasmodium* metabolic processes.

In experimental procedures, clarify whether you utilized isolated fat bodies or abdominal walls with adhered fat bodies (a common practice for a culturing of the mosquito fat body).

Reference list should be checked throughout. At the first glance, I noticed that in the first reference Ahmed A. is listed as Ahmed a.; Bryant et al (2011) should be just Bryant and Raikhel (2011).

Reviewer #2:

Remarks to the Author:

This study reported the roles of miR276-mediated metabolic balancing in reproductive cycle and *Plasmodium* development in mosquitoes. Genetic manipulation of miR-276 prolonged amino acid catabolism and enhanced mosquito fertility. Prolongation of the reproductive period in *Plasmodium* infected females impaired the parasitic development. Overall, the scientific conceptual raised by this study, which is the metabolic interaction in reproductive cycle and *Plasmodium* development after mosquito blood meal, is interesting. Nonetheless, the miR276 and BCAT-related phenotypes were

really weak (Figure 1A; Figure 2A, C, D; Figure 4). My major concern is that the miR276/BCAT might not be the dominant factors in regulation of this metabolic balance.

1. Question to the rationale of this study. Line 87-91, "However, how the miRNAs that regulate mosquito metabolism impact vector-parasite interactions is unknown. In *A. gambiae*, a major malaria vector in sub-Saharan Africa, blood feeding increases the fat body levels of miR-276-5p transcripts (miR-276 hereafter) during the mid-/late-phase of the reproductive cycle (Lampe and Levashina, 2018)." Blood meal may regulate expression of many small RNA in mosquitoes. Why did they choose microRNA? I believe that hundreds of mosquito genes/small RNAs can be dramatically regulated by blood meal. Is miR-276 the most dramatically regulated miRNA after blood meal?

2. Expression of miRNA-276 was just enhanced around 2 folds by blood meal (with large variation among individuals). How does miRNA-276 play an important role in regulation of reproductive cycle and Plasmodium development? I question that miR276 might not be the dominant factor in the process.

3. As the authors mentioned in the manuscript, mRNA regulation was not globally measured in the miR-276-manipulated mosquitoes. They cannot exclude that miR-276 regulates other important genes more than BCAT. Obviously, the authors realized the weakness here, why they did not do genome-wide analysis to give more evidence in this study.

4. Silencing miR-276 increases the mosquito fertility however inhibits Plasmodium sporogonic development, the phenomenon is interesting. What is the mechanism to explain this phenotype?

Reviewer #3:

Remarks to the Author:

The paper claims that miR-276 controls important metabolite changes that influences the reproductive cycle of mosquitoes. miR-276 inhibits amino acid catabolism so that nutrients can be used for mosquito egg development. The Plasmodium parasites thrives in tandem with the nutrients available and thus is influenced by miR-276.

The work is clear and convincing as it makes use of metabolomics to show the metabolic changes with miR-276KD when compared to control mosquitoes. Also, eggs and sporogonic index were shown to be influenced by miR-276KD, which further confirmed how the different factors are inter-related.

Results of the manuscript will be of interest to researchers in the field.

Comments on the metabolomics part of the work:

1. It is stated that metabolomics analysis was performed by a commercial company, MetaSysX. Other than that, there was insufficient information for the work to be fully reproduced. For example, sample preparation was stated to be modified from Giavalisco et al. However sample preparation or extraction procedure was absent in the cited paper, which itself had referenced to another publication. In addition, LC-MS running protocol was not stated nor cited, while GC-MS running protocol was stated in this manuscript. This should be standardised.

2. In methods section, there was a section on MS/MS lipid annotation, but there was no description of lipids from the metabolomics analysis in the results or discussion. Recommendation would be to remove this section from the methods section if no relevant lipid analysis results was discussed.

3. Discussion of the heat map (Fig 3A) can be much improved. In its discussion (line 230 to 239), appearance and disappearance of clusters of metabolites correlated with blood digestion and different

phases of the reproductive cycles can be further described, accompanied by the identities of metabolites detected. Without information of the metabolites, changes can be simply due to the blood being digested (disappearance of blood compounds/nutrients) or the increase in egg development (such as lipids?). Metabolite identification can help make useful inferences such as hormonal changes and other changes in metabolism. In line 575, it was stated that a library of reference compound was used, thus some identification could have been made.

4. Most of the results were made by studying the changes in amino acid levels. Comparisons of amino acid levels were done by investigating fold changes of metabolites of miR-276KD with scramble. Within miR-276KD and scramble samples, how were the sample results normalised? Ideally they should be normalised by wet weight or total protein, not by number of mosquitoes. The normalisation factor can have a huge effect on the accuracy of metabolomics results.

Minor comments:

Line 231: "At the early as 10hpb..." change to "As early as 10hpb"?

Jianhong Ching, PhD

Reviewer #1 (Remarks to the Author):

1. It is important to add wild type controls in most of physiological experiments and profile measuring, for example in Fig. 2D and 4. Utilization of the miR-276 scrambled probe is essential. It is, however, could be misleading because of a potential adverse effect of injection or potential effects of a foreign molecule introduction.

Analyses of miRNA function was performed by injection of a miR-276 antagomir. Previous studies showed that injection of PBS or double-stranded RNA reduces levels of *Plasmodium falciparum* in the mosquito by a wounding-induced AP-1/FOS-TGase2 regulatory axis^{1,2}. In line with this report, we observed a pronounced effect of control (scramble) antagomir injection on *Plasmodium falciparum* development (Supplementary Figure 5). Because of this observation, we considered injection of a scrambled miR-276 antagomir as the most appropriate control for the anti-miR-276 antagomir. This type of control is widely accepted in miRNA studies across diverse fields⁽³⁻⁶⁾, and in RNA interference (RNAi) experiments (Blandin et al., 2004; Bryant and Raikhel, 2011; Gabrieli et al., 2014; Roy et al., 2015; Shaw et al., 2014 to cite a few).

2. Measuring the effect of miR-276 on the postblood meal (PBM) phase should also be evaluated by such an important parameter as the duration of the Vg expression period and not only egg number, which does not show actual duration of this process. This will be a more accurate indicator of the delay in the cycle.

We thank the reviewer for this comment and apologize for the confusion. Our hypothesis was not based on egg numbers, we agree that this phenotype is not a good marker to estimate the duration of the reproductive cycle. We formulated our hypothesis based on the observed changes in BCAT transcript levels in control and miR-276-silenced mosquitoes. The prolongation of high levels of *BCAT* expression was consistent with the previous reports in *Aedes*, that demonstrated the key role of the switch from an anabolic to catabolic metabolism in the fat body during the post-blood meal phase^{12,13}. As BCAT performs the first step of the BCAA catabolism, high levels of its expression can be considered as a marker of the catabolic phase of the fat body metabolism. However, we agree that “prolongation of the reproductive cycle” is a very general term and rephrased our wording to “prolongation of high rates of AA catabolism.

Furthermore, we followed the advice of the reviewer and provided additional data to support the correlation between the duration and intensity of the AA catabolic phase and the duration of the reproductive cycle by using *Vitellogenin* as a readout. Vg is a yolk protein which is produced and secreted by the mosquito fat body after blood feeding and its transcript levels plunge to zero at 48 h post blood feeding¹⁴⁻¹⁶. We examined expression of *Vitellogenin* (*Vg*) mRNA levels at 38 h post blood meal in control and miR-276 silenced mosquitoes. We show that similarly to *BCAT*, silencing of miR-276 prolongs expression of *Vg* (Supplementary Figure 6). This additional data extends the impact of miR-276 manipulation beyond the effect on AA catabolism to lipid metabolism. We have introduced the following paragraph into the revised manuscript: page 20 line 386:

“These conclusions are further supported by the observed changes in expression of the major yolk protein gene *Vitellogenin (Vg)* (Supplementary Figure 6). As *Vg* lacks miR-276 binding sites in its 3’UTR and cannot be directly regulated by this miRNA, the prolongation of the AA catabolism must be coordinated with lipid metabolism, an interesting observation which calls for further investigation.”

3. It is essential to utilize mimic phenotype rescue experiments. Will the reproductive cycle be restored following miR-276 antagomir and mimic applications; what is the mimic effect on the target enzyme expression; effect on the timing of vitellogenic period and egg number; finally, on Plasmodium development.

While we agree with the referee that restoring miR-276 function in the knockdown context would have been ideal to establish causality, phenotype rescue experiments are not possible in the current system in the absence of a mutant miR-276^{-/-} mosquito line. Simultaneous injection of miR-276 antagomir and mimic would result in binding of the miRNA mimic to the antagomir and prevent meaningful conclusions. Moreover, female mosquitoes can only be injected before blood feeding as *P. falciparum*-infected mosquitoes are highly fragile and have to be manipulated in S3 conditions. Injection of miR-276 mimic before blood feeding may cause broad, possibly irrelevant, phenotypes, as it will lack the crucial temporal resolution. Additionally, high levels of miR-276 mimic may favour off-target effects. Similarly, a rescue of miR-276 silencing phenotype by knockdown of *BCAT* would require injection of big quantities of antagomir and dsRNA and lack temporal regulation. Nevertheless, to address the reviewer’s concern, we performed additional experiments to confirm the role of *BCAT* in mosquito egg development using a direct approach (Supplementary Figure 4). Our results show that knockdown of *BCAT* during the reproductive cycle decreases egg development. This result is in accordance to the reported phenotype of increased egg development caused by prolonged *BCAT* expression in miR-276^{KD} mosquitoes. The following sentence was introduced into the text: page 16 line 289.

“To provide further evidence that *BCAT* is the target of miR-276, we examined the effect of *BCAT* knockdown on oogenesis. In line with the results obtained for miR-276^{KD}, *BCAT* silencing by injection of dsRNA prior to blood feeding decreased egg laying, suggesting a critical role of amino acid catabolism in mosquito reproduction (Supplementary Figure 4).”

4. Fig. 1 needs an explanation. The authors have shown that amino acids and 20-hydroxyecdysone (20E) work together in activation of miR-276 expression. Such regulation has been demonstrated for a number of mosquito genes that are expressed during the first half of the PBM phase (about 6-30h). However, miR-276 expression extends at its high level until 44 hr, way after the 20E titer declines to the background level (36 hr PBM). Thus, the regulation of the miR-276 expression is more complex than it has been presented here. To more thoroughly investigate miR-276 regulation, it would be advisable to investigate whether the effect of 20E on miR-276 is direct through its receptor EcR. Analysis of the miR-276 gene promoter for potential binding sites for EcR or its early gene E74 could be done here.

We used an *ex vivo* fat body cultures to investigate the role of amino acids and 20-hydroxyecdysone in the regulation miR-276 expression. Our results showed that both signals are required to achieve the highest miR-276 transcript levels. This important experimental evidence suggests that similarly to miR-275 and other miRNAs, expression of miR-276 is induced either directly or indirectly by AAs and 20E from 24 to 36h post blood feeding (see also *bantam*)¹⁷. A previous study showed that miRNAs have better stability and a 2-20-fold longer half-life compared to mRNAs *in vivo*¹⁸. Therefore, we believe that while induction of miR-276 could rely on similar transcriptional regulation, the stability and/or half-life of miR-276 may differ from other protein-encoding transcripts and explain the observed phenomenon.

Following the reviewer's advice, we investigated the potential transcription factor binding sites in the 1.2 kB upstream region of the miR-276 gene. While no EcR binding sites were identified, we found a potential binding site for Broad transcription factor. *Broad* is a direct target of the ecdysone receptor¹⁹ and may be the miR-276 inducing factor. We introduced this finding into the text: Page 7 line 119:

"In line with these results, bioinformatics analysis identified one potential binding site for the ecdysone-induced transcriptional factor Broad in the upstream regulatory region of the miR-276 gene."

5. I find a part of study investigating the effect of miR-276 on Plasmodium development to be rather superficial. For example, one cannot exclude a direct effect of miR-276 on Plasmodium metabolic processes.

We disagree with the reviewer about the superficial nature of our *P. falciparum* infections. These experiments involved work in S3 conditions and were not easy to perform. In contrast to the majority of published studies that focus on oocyst numbers, we followed parasite development to its final stage in the mosquito, the sporozoites. We report 6 results of 6 independent experiments and describe a complex phenotype of parasite development that involves not only the number but also the size of oocysts. Therefore, we introduce a very informative measure, the sporogonic index, that gives a cumulative reading of these two variables. As for the referee's concern about the direct effect of the mosquito miR-276 on *P. falciparum* metabolic processes, it is known that *Plasmodium* parasites lack miRNAs as well as the molecular machinery including Dicer proteins^{20,21}. It is also unlikely, that miRNA synthesized in the mosquito fat body cells will penetrate the oocyst wall and taken up by the parasites. On the other hand, the antagomirs directed against mosquito miRNA do not have miRNA targets in the parasites²⁰⁻²². Off-target effects on mosquito and parasite cells cannot be completely excluded as in all other RNAi techniques. As we injected antagomirs four days before mosquito infection with *P. falciparum*, the vast majority of the circulating antagomir molecules should have been taken up or cleared up by the mosquito cells.

In experimental procedures, clarify whether you utilized isolated fat bodies or abdominal walls with adhered fat bodies (a common practice for a culturing of the mosquito fat body).

We added the requested information to the experimental procedures and the result section: Page 6 line 97:

“To define the role of miR-276 in the mosquito reproductive cycle, we examined its expressional profile by reverse transcription quantitative real-time PCR (RT-qPCR) in the abdominal carcass tissues collected before and after blood feeding. As the fat body is highly enriched in abdominal carcasses, we refer to them as fat body samples hereafter.”

And method section: Page 25 line 478:

“The abdominal carcasses (the fat body) were immediately homogenized in RNAzol (Sigma Aldrich) and kept at -80°C for RNA isolation.”

Reference list should be checked throughout. At the first glance, I noticed that in the first reference Ahmed A. is listed as Ahmed a.; Bryant et al (2011) should be just Bryant and Raikhel (2011).

We thank the reviewer for this comment and modified the references accordingly.

Reviewer #2 (Remarks to the Author):

Overall, the scientific conceptual raised by this study, which is the metabolic interaction in reproductive cycle and Plasmodium development after mosquito blood meal, is interesting. Nonetheless, the miR276 and BCAT-related phenotypes were really weak (Figure 1A; Figure 2A, C, D; Figure 4). My major concern is that the miR276/BCAT might not be the dominant factors in regulation of this metabolic balance.

1. Question to the rationale of this study. Line 87-91, "However, how the miRNAs that regulate mosquito metabolism impact vector-parasite interactions is unknown. In *A. gambiae*, a major malaria vector in sub-Saharan Africa, blood feeding increases the fat body levels of miR-276-5p transcripts (miR-276 hereafter) during the mid-/late-phase of the reproductive cycle (Lampe and Levashina, 2018)." Blood meal may regulate expression of many small RNA in mosquitoes. Why did they choose microRNA? I believe that hundreds of mosquito genes/small RNAs can be dramatically regulated by blood meal. Is miR-276 the most dramatically regulated miRNA after blood meal?

We thank the reviewer for this comment and reformulated the text to make the rationale of this study clearer: Page 4 line 79:

"MicroRNAs contribute to post-transcriptional regulation and link the endocrine regulation with metabolic homeostasis in *Aedes* mosquitoes²²⁻²⁴. Using a transcriptomic approach in *A. gambiae*, we identified three miRNAs whose fat body expression was induced shortly after blood feeding, namely miR-275, miR-276, and miR-305²⁵. Consistently with our findings, miR-275 was reported to regulate blood meal digestion and egg development²⁴, while miR-305 was shown to impact mosquito microbiota and *P. falciparum* development by an as yet unknown mechanism²⁶. Here, we report the role of miR-276 in fine-tuning the expression of *branched chain amino acid transferase (BCAT)* in the fat body. We show that miR-276 depletion prolongs high levels of *BCAT* expression and AA catabolism in the fat body, thereby increasing female fertility. Unexpectedly, sustained high levels of AA catabolism compromised *P. falciparum* sporogonic development and reduced the number of transmissible sporozoites. Our results demonstrate the important role of mosquito metabolism in vector competence and malaria transmission."

2. Expression of miRNA-276 was just enhanced around 2 folds by blood meal (with large variation among individuals). How does miRNA-276 play an important role in regulation of reproductive cycle and Plasmodium development? I question that miR276 might not be the dominant factor in the process.

The blood meal-induced induction of miR-276 in the fat body varies between 2- to 4-fold depending on the biological replicate. Such variation in transcript levels may result from heterogeneity in timing of miR-276 expression between individual mosquitoes and independent replicates during the six-hour sampling interval. Furthermore, 2-4-fold miRNA induction by blood feeding is in line with the previous reports on other mosquito miRNAs whose expression is regulated by blood feeding. For example, while levels of miR-275 transcripts were only 7-fold induced, this miRNA was crucial for blood meal digestion¹⁷. Similarly, in *Aedes* mosquitoes, blood

meal induced 3-5-fold changes in the expression of the major blood meal-regulated miR-79, miR-100 and miR-11¹⁷. Note that in other models, 2-fold upregulation of miR-150 decreased levels of adiponectin receptor 2 and cell stress resistance²⁴. As miRNAs tune mRNA levels, they do not require high levels of upregulation to limit the expression of targeted mRNAs. In fact, the 2- to 4-fold upregulation in miR-276 transcript levels causes 50% reduction in BCAT levels at 38 hpb, quite an impressive phenotype.

We would also like to point out that we do not claim that miR-276 is the “dominant factor of the reproductive cycle and *Plasmodium* development”. We propose that this miRNA fine-tunes the mosquito reproductive cycle, especially the transition between the catabolic and anabolic stages (please see the title and conclusions of the manuscript). However, the balancing function of miR-276 in regulating mosquito metabolism does not diminish the importance of our findings. On the contrary, it stresses the importance of metabolic homeostasis in mosquito reproduction and parasite development. Knockdown of crucial metabolic factors during the reproductive cycle often has lethal consequences^{25–28} and precludes further understanding of intricate relationships between ovary and parasite development. Here, we show that miR-276 fine-tunes the switch from a catabolic to anabolic amino acid metabolism in the fat body, and that interfering with this fine-tuning decreases the allocation of reproductive investment by 30% and impedes parasite sporogony. The fine regulation of reproductive investment is likely even more important for field mosquitoes, for which resource allocation is crucial for survival.

3. As the authors mentioned in the manuscript, mRNA regulation was not globally measured in the miR-276-manipulated mosquitoes. They cannot exclude that miR-276 regulates other important genes more than BCAT. Obviously, the authors realized the weakness here, why they did not do genome-wide analysis to give more evidence in this study.

We agree with the referee that we cannot exclude that miR-276 could have multiple targets in the mosquito, we specifically stated this limitation in the text. However, we decided to focus our study on one of the miR-276 targets. BCAT is the first miRNA target identified in *A. gambiae* to date, and we thought it was important to communicate these findings. Furthermore, we now provide the additional data that further support the role of the miR-276 – BCAT axis in the regulation of expression of the yolk protein Vitellogenin (Supplementary Figure 6) and ovary development (see our answer 2 to the reviewer 1 and Supplementary Figure 4).

4. Silencing miR-276 increases the mosquito fertility however inhibits Plasmodium sporogonic development, the phenomenon is interesting. What is the mechanism to explain this phenotype?

A large proportion of amino acids are catabolized in the mosquito fat body for ATP production²⁹. This demand for ATP sustains the high metabolic activity of the fat body after blood feeding and provides energy for yolk protein production and nutrient mobilization. Knockdown of the miR-276 leads to an increased and prolonged expression of *BCAT*, the enzyme that regulates the first step of AA catabolism, thereby promoting synthesis of yolk proteins and mobilization of lipids and glycogen.

In line with this hypothesis, miR-276-silencing increases expression levels of the yolk protein Vitellogenin at 38 hpb (Supplementary Figure 6). The amino acid catabolism is an important regulator of mosquito egg development^{30,31}. Here, we show that the increased levels of BCAT induced by miR-276 silencing, benefit mosquito fertility (Figure 4A). In contrast, knockdown of *BCAT* results in lower fertility (Supplementary Figure 4). During the first gonotrophic cycle egg development and parasite proliferation are decoupled in time^{32,33}. We propose that higher levels of the mosquito reproductive investment consume nutrient resources and, thereby, restrict sporogonic development of the parasites. We provide a detailed explanation of the underlying mechanism in the discussion and Figure 5.

Reviewer #3 (Remarks to the Author):

The paper claims that miR-276 controls important metabolite changes that influences the reproductive cycle of mosquitoes. miR-276 inhibits amino acid catabolism so that nutrients can be used for mosquito egg development. The Plasmodium parasites thrives in tandem with the nutrients available and thus is influenced by miR-276.

The work is clear and convincing as it makes use of metabolomics to show the metabolic changes with miR-276KD when compared to control mosquitoes. Also, eggs and sporogonic index were shown to be influenced by miR-276KD, which further confirmed how the different factors are inter-related.

Results of the manuscript will be of interest to researchers in the field.

Comments on the metabolomics part of the work:

1. It is stated that metabolomics analysis was performed by a commercial company, MetaSysX. Other than that, there was insufficient information for the work to be fully reproduced. For example, sample preparation was stated to be modified from Giavalisco et al. However, sample preparation or extraction procedure was absent in the cited paper, which itself had referenced to another publication. In addition, LC-MS running protocol was not stated nor cited, while GC-MS running protocol was stated in this manuscript. This should be standardised.

We thank the reviewer for this comment on the metabolomics method section. We now provide a more detailed material and methods. We also updated the citation for the sample preparation and the LC-MS protocol. We added this information to the Materials and Methods section of the manuscript on page 28 line 547:

“Sample preparation

The sample preparation was performed according to MetaSysX standard procedure, a modified protocol from Salem *et al.*⁵⁴. Whole bodies of snap-frozen mosquitoes were grounded into a homogenous fine powder using a tissue homogenizer and equal amounts for aliquoted and used for untargeted metabolomics carried out by MetaSysX, Potsdam, Germany.

LC-MS measurements (hydrophilic and lipophilic analytes)

The samples were measured with a Waters ACQUITY Reversed Phase Ultra Performance Liquid Chromatography (RP-UPLC) coupled to a Thermo-Fisher Q-Exactive mass spectrometer⁵⁴. C8 and C18 columns were used for the lipophilic and the hydrophilic measurements, respectively. Chromatograms were recorded in Full Scan MS mode (Mass Range [100-1500]).”

2. In methods section, there was a section on MS/MS lipid annotation, but there was no description of lipids from the metabolomics analysis in the results or discussion.

Recommendation would be to remove this section from the methods section if no relevant lipid analysis results was discussed.

We included the lipid analysis as the heatmap and supplementary data file includes GC- and LC-MS data. We hope that this comprehensive dataset can be of use to other researchers and therefore we provide the full set of material and methods for both GC- and LC-MS data.

3. Discussion of the heat map (Fig 3A) can be much improved. In its discussion (line 230 to 239), appearance and disappearance of clusters of metabolites correlated with blood digestion and different phases of the reproductive cycles can be further described, accompanied by the identities of metabolites detected. Without information of the metabolites, changes can be simply due to the blood being digested (disappearance of blood compounds/nutrients) or the increase in egg development (such as lipids?). Metabolite identification can help make useful inferences such as hormonal changes and other changes in metabolism. In line 575, it was stated that a library of reference compound was used, thus some identification could have been made.

We thank the reviewer for this suggestion. Indeed, in total 816 metabolites were annotated using the mentioned reference library. We included the metabolite annotation in the supplementary data file comprising the GC- and LC-MS data.

To make overall metabolite changes more informative, we focused on the annotated features and visualized metabolites that displayed significant changes across the reproductive cycle in Figure 3A. Using the ward clustering method, three distinct clusters were annotated in Figure 3A and the within clusters metabolite annotations were added as Supplementary Table 4. Furthermore, we introduced to the results a more detailed description of the metabolic changes across the reproductive cycle: Page 12 line 225

“Massive metabolic changes after blood feeding were observed in both control and miR-276^{KD} mosquitoes (Figure 3A and Supplementary Table 4). As early as 10 hpb, a large cluster of highly enriched metabolites correlated with the influx of human blood (Cluster I). This cluster comprised amino acids, short and long chain fatty acids, as well as sphingomyelins, potentially representing metabolites derived from the ingested blood. A second cluster (Cluster II), comprising predominantly dipeptides, was abundant at 10 and 38 hpb and disappeared at the late phase of the reproductive cycle (48 hpb). Conversely, Cluster III featured the metabolites whose levels increased at 38 and 48 hpb, corresponding to the periods of blood digestion and ovary development. The vast majority of Cluster III metabolites were lipids (triacylglycerides (TAGs), phosphatidylcholines, phosphatidylethanolamines, phosphatidylinositols) and trehalose (Figure 3A and Supplementary Table 4). Overall, the observed metabolite dynamics was not significantly perturbed by miR-276^{KD} and closely recapitulated the major physiological processes induced by blood feeding. This observation was in line with the fine-tuning role of miR-276, whose silencing did not hinder the overall blood feeding-induced metabolic profile.”

4. Most of the results were made by studying the changes in amino acid levels.

Comparisons of amino acid levels were done by investigating fold changes of metabolites of miR-276KD with scramble. Within miR-276KD and scramble samples, how were the sample results normalised? Ideally, they should be normalised by wet weight or total protein, not by number of mosquitoes. The normalisation factor can have a huge effect on the accuracy of metabolomics results.

Ten mosquitoes were collected and snap frozen per sample and time point. All collected samples were reduced to fine powder and equal amounts of dried powder were used for further downstream GC- and LC-MS. The raw data was then normalized to sample median.

Minor comments:

Line 231: "At the early as 10hpb..." change to "As early as 10hpb"?

The text was changed according to the reviewer's suggestion.

Jianhong Ching, PhD

References:

1. Nsango, S. E. *et al.* Genetic clonality of *Plasmodium falciparum* affects the outcome of infection in *Anopheles gambiae*. *International journal for parasitology* **42**, 589–95 (2012).
2. Nsango, S. E. *et al.* AP-1/Fos-TGase2 axis mediates wounding-induced *Plasmodium falciparum* killing in *Anopheles gambiae*. *The Journal of biological chemistry* **288**, 16145–54 (2013).
3. Dennison, N. J., BenMarzouk-Hidalgo, O. J. & Dimopoulos, G. MicroRNA-regulation of *Anopheles gambiae* immunity to *Plasmodium falciparum* infection and midgut microbiota. *Developmental and comparative immunology* **49**, 170–8 (2015).
4. Maschmeyer, P. *et al.* Selective targeting of pro-inflammatory Th1 cells by microRNA-148a-specific antagomirs in vivo. *Journal of autoimmunity* **89**, 41–52 (2018).
5. Xu, L.-J., Ouyang, Y.-B., Xiong, X., Stary, C. M. & Giffard, R. G. Post-stroke treatment with miR-181 antagomir reduces injury and improves long-term behavioral recovery in mice after focal cerebral ischemia. *Experimental neurology* **264**, 1–7 (2015).
6. Yang, Y. *et al.* MiR-135 suppresses glycolysis and promotes pancreatic cancer cell adaptation to metabolic stress by targeting phosphofructokinase-1. *Nature communications* **10**, 809 (2019).
7. Roy, S. *et al.* Regulation of Gene Expression Patterns in Mosquito Reproduction. *PLoS genetics* **11**, e1005450 (2015).
8. Bryant, B. & Raikhel, A. S. Programmed Autophagy in the Fat Body of *Aedes aegypti* Is Required to Maintain Egg Maturation Cycles. *PLoS ONE* **6**, e25502 (2011).
9. Shaw, W. R. *et al.* Mating activates the heme peroxidase HPX15 in the sperm storage organ to ensure fertility in *Anopheles gambiae*. *Proceedings of the National Academy of Sciences of the United States of America* **111**, 5854–9 (2014).
10. Blandin, S. *et al.* Complement-Like Protein TEP1 Is a Determinant of Vectorial Capacity in the Malaria Vector *Anopheles gambiae*. *Cell* **116**, 661–670 (2004).
11. Gabrieli, P. *et al.* Sexual transfer of the steroid hormone 20E induces the postmating switch in *Anopheles gambiae*. *Proceedings of the National Academy of Sciences* **111**, 16353–16358 (2014).
12. Hou, Y. *et al.* Temporal Coordination of Carbohydrate Metabolism during Mosquito Reproduction. *PLoS genetics* **11**, e1005309 (2015).
13. Wang, X. *et al.* Hormone and receptor interplay in the regulation of mosquito lipid metabolism. *Proceedings of the National Academy of Sciences of the*

United States of America 201619326 (2017). doi:10.1073/pnas.1619326114

14. Marinotti, O., Capurro, M. de L., Nirmala, X., Calvo, E. & James, A. A. Structure and expression of the lipophorin-encoding gene of the malaria vector, *Anopheles gambiae*. *Comparative biochemistry and physiology. Part B, Biochemistry & molecular biology* **144**, 101–9 (2006).
15. Nirmala, X., Marinotti, O. & James, A. A. The accumulation of specific mRNAs following multiple blood meals in *Anopheles gambiae*. *Insect Molecular Biology* **14**, 95–103 (2005).
16. Rono, M. K., Whitten, M. M. A., Oulad-Abdelghani, M., Levashina, E. A. & Marois, E. The major yolk protein vitellogenin interferes with the anti-plasmodium response in the malaria mosquito *Anopheles gambiae*. *PLoS biology* **8**, e1000434 (2010).
17. Bryant, B., Macdonald, W. & Raikhel, A. S. microRNA miR-275 is indispensable for blood digestion and egg development in the mosquito *Aedes aegypti*. *Proceedings of the National Academy of Sciences of the United States of America* **107**, 22391–8 (2010).
18. Zhang, Z., Qin, Y.-W., Brewer, G. & Jing, Q. MicroRNA degradation and turnover: regulating the regulators. *Wiley interdisciplinary reviews. RNA* **3**, 593–600 (2012).
19. Chen, L., Zhu, J., Sun, G. & Raikhel, A. S. The early gene Broad is involved in the ecdysteroid hierarchy governing vitellogenesis of the mosquito *Aedes aegypti*. *Journal of molecular endocrinology* **33**, 743–61 (2004).
20. Xue, X., Zhang, Q., Huang, Y., Feng, L. & Pan, W. No miRNA were found in Plasmodium and the ones identified in erythrocytes could not be correlated with infection. *Malaria journal* **7**, 47 (2008).
21. Baum, J. *et al.* Molecular genetics and comparative genomics reveal RNAi is not functional in malaria parasites. *Nucleic acids research* **37**, 3788–98 (2009).
22. Rathjen, T., Nicol, C., McConkey, G. & Dalmay, T. Analysis of short RNAs in the malaria parasite and its red blood cell host. *FEBS Letters* **580**, 5185–5188 (2006).
23. Lampe, L. & Levashina, E. A. MicroRNA Tissue Atlas of the Malaria Mosquito *Anopheles gambiae*. *G3 (Bethesda, Md.)* **8**, 185–193 (2018).
24. Kreth, S. *et al.* MicroRNA-150 inhibits expression of adiponectin receptor 2 and is a potential therapeutic target in patients with chronic heart failure. *J Heart Lung Transplant* **33**, 252–260 (2014).
25. Sterkel, M. *et al.* Tyrosine Detoxification Is an Essential Trait in the Life History of Blood-Feeding Arthropods. *Current Biology* **26**, 2188–2193 (2016).
26. Sterkel, M., Oliveira, J. H. M., Bottino-Rojas, V., Paiva-Silva, G. O. & Oliveira, P. L. The Dose Makes the Poison: Nutritional Overload Determines the Life

Traits of Blood-Feeding Arthropods. *Trends in Parasitology* **33**, 633–644 (2017).

27. Isoe, J. *et al.* Xanthine dehydrogenase-1 silencing in *Aedes aegypti* mosquitoes promotes a blood feeding–induced adulticidal activity. *The FASEB Journal* **31**, 2276–2286 (2017).
28. Magalhaes, T., Brackney, D. E., Beier, J. C. & Foy, B. D. Silencing an *Anopheles gambiae* catalase and sulfhydryl oxidase increases mosquito mortality after a blood meal. *Archives of insect biochemistry and physiology* **68**, 134–43 (2008).
29. Zhou, G., Pennington, J. E. & Wells, M. A. Utilization of pre-existing energy stores of female *Aedes aegypti* mosquitoes during the first gonotrophic cycle. *Insect Biochemistry and Molecular Biology* **34**, 919–925 (2004).
30. Mazzalupo, S., Isoe, J., Belloni, V. & Scaraffia, P. Y. Effective disposal of nitrogen waste in blood-fed *Aedes aegypti* mosquitoes requires alanine aminotransferase. *FASEB journal : official publication of the Federation of American Societies for Experimental Biology* **30**, 111–20 (2016).
31. Fuchs, S. *et al.* Phenylalanine Metabolism Regulates Reproduction and Parasite Melanization in the Malaria Mosquito. *PLoS ONE* **9**, e84865 (2014).
32. Costa, G. *et al.* Non-competitive resource exploitation within mosquito shapes within-host malaria infectivity and virulence. *Nature Communications* **9**, 3474 (2018).
33. Werling, K. *et al.* Steroid Hormone Function Controls Non-competitive Plasmodium Development in *Anopheles*. *Cell* **177**, 315-325.e14 (2019).
34. Scaraffia, P. Y. *Disruption of Mosquito Blood Meal Protein Metabolism. Genetic Control of Malaria and Dengue* (Elsevier Inc., 2015).
doi:10.1016/B978-0-12-800246-9.00012-0

Reviewers' Comments:

Reviewer #1:

Remarks to the Author:

I find that the authors have satisfactorily addressed my concerns.

Reviewer #2:

Remarks to the Author:

No more comment

Reviewer #3:

Remarks to the Author:

I thank the authors for having fully answered my queries on the metabolomics section.

To the response of including Supplementary Table 4, it would be appropriate for the authors to include in the table the percentage matching scores of the metabolite annotation according to one or more specific chemical libraries (online, in-house, or otherwise). This is important for exhibiting confidence of the untargeted metabolomic techniques in annotating metabolites. Low matching scores compromise on data accuracy, and can have a huge impact on the results. Typical acceptable matching scores would be 90 and above percent before they can be used in discussions. Such information can also be described in the methods section under LC-MS data annotation.

For essential results on the amino acids, it would be best if the authors have validated the metabolite identities with corresponding authentic standards, especially if the standards are easily available commercially.

Jianhong Ching

Answers to reviewer 3

We thank the referee for the constructive comments on the metabolomics aspect of our manuscript and agree that metabolomics data should be presented according to the highest possible standards. We are currently witnessing important changes in standards that regulate reporting of metabolic methods, data description and sharing. As our analyses were outsourced to the MetaSysX company three years ago, the contract agreement which was signed with the company at that time did not include raw data sharing. Given the substantial amount of work requested by the reviewer, we invited the company scientist involved in our project as an additional co-author. All authors agreed to this inclusion. We are now providing the additional information on the data quality and interpretation performed by MetaSysX. Please find below our point-by-point answers.

1. *"I thank the authors for having fully answered my queries on the metabolomics section.*

To the response of including Supplementary Table 4, it would be appropriate for the authors to include in the table the percentage matching scores of the metabolite annotation according to one or more specific chemical libraries (online, in-house, or otherwise). This is important for exhibiting confidence of the untargeted metabolomic techniques in annotating metabolites. Low matching scores compromise on data accuracy, and can have a huge impact on the results. Typical acceptable matching scores would be 90 and above percent before they can be used in discussions. Such information can also be described in the methods section under LC-MS data annotation."

The company used the in-house IP-protected algorithm and library to annotate the mass peaks from untargeted LC-MS. This algorithm does not calculate a percentage matching score but uses other criteria for compound annotation. Note that the percentage matching scores were only recently introduced as QC and that many recently published high impact metabolomics papers do not report them¹⁻⁵. The company follows a good practice of metabolite annotation and provided now a detailed description of the annotation pipeline.

The metaSysX library was measured under the same chromatographic and spectrometric conditions as the experimental sample measurements. The database included retention time and high-resolution m/z information of detected ions from measurements of authentic standards. In general, compounds were annotated based on their retention time, mass, fragmentation pattern and elution patterns. The following cut-offs were used for compound annotation: for polar platforms +/- 5 ppm and +/-0.095 min deviation from the reference compounds m/z and retention time, respectively, were used as matching criteria; for lipid annotation +/- 5 ppm and the

range -0.095, +0.255 min time deviation. In many cases, the annotation was supported by detection of more than one ion fragment representing the same analyte. Furthermore, experimental samples were pooled and independently measured four times using LC-MS/MS mode. The fragmentation was performed in data-dependent acquisition mode using higher-energy collisional dissociation (HCD). The lipid fragmentation data was used to increase the confidence of the metabolite annotations. The in-house created R-based annotation algorithm used information from fragments, neutral losses and precursor mass information detected in both polarities. Any coeluting compounds with the same parent ion mass were not annotated.

The compound RT and mass deviation from the reference metabolite is reported in the Supplementary data file 1. Furthermore, we included the mass accuracy and LC-MS gradient length to the Material and Methods section to provide the context of the above-mentioned values. Amino acids, the main focus of this manuscript, were often detected in more than one platform (positive/negative mode LC-MS and/or GC-MS) and therefore the confidence of this annotation is very high.

We now provide a more detailed description of the sample preparation, extraction, LC-MS method and annotation in Material and Methods section (see page 28, line 542).

2. *“For essential results on the amino acids, it would be best if the authors have validated the metabolite identities with corresponding authentic standards, especially if the standards are easily available commercially.”*

The MetaSysX database includes retention time and the m/z information of detected ions from measurements of authentic standards using the same instrument and analysis conditions. Therefore, compounds have been validated against authentic standards. We added the following statement to emphasize this aspect within the text:

Page 12, line 222:

In total, we detected 4,716 chromatographic peaks (*m/z* at a specific retention time) of which 816 were annotated in reference of authentic standards.

References

1. Nair, S. *et al.* Adult stem cell deficits drive Slc29a3 disorders in mice. *Nature Communications* **10**, 2943 (2019).
2. Tabassum, R. *et al.* Genetic architecture of human plasma lipidome and its link to cardiovascular disease. *Nature Communications* **10**, 4329 (2019).

3. Goultquer, S. *et al.* Consequences of blunting the mevalonate pathway in cancer identified by a pluri-omics approach. *Cell Death & Disease* **9**, 745 (2018).
4. Veglia, F. *et al.* Fatty acid transport protein 2 reprograms neutrophils in cancer. *Nature* **569**, 73–78 (2019).
5. Parker, B. L. *et al.* An integrative systems genetic analysis of mammalian lipid metabolism. *Nature* **567**, 187–193 (2019).